# A novel framework for increasing research transparency: Exploring the connection between diversity and innovation

Timothy R. Wojan[1]*, Dayton M. Lambert[2]

**1** Oak Ridge Institute for Science and Education Research Ambassadors Program, National Center for Science and Engineering Statistics, U.S. National Science Foundation, Alexandria, Virginia, United States of America, **2** Department of Agricultural Economics, Oklahoma State University, Stillwater, Oklahoma, United States of America

* twojan@nsf.gov

**Data Availability Statement:** The Census Bureau/National Center for Science and Engineering Statistics data used in the analysis were collected under pledges of confidentiality as stipulated in Title 13 and Title 26. These data are only available

## Abstract

A split sample/dual method research protocol is demonstrated to increase transparency while reducing the probability of false discovery. We apply the protocol to examine whether diversity in ownership teams increases or decreases the likelihood of a firm reporting a novel innovation using data from the 2018 United States Census Bureau's Annual Business Survey. Transparency is increased in three ways: 1) all specification testing and identifying potentially productive models is done in an exploratory subsample that 2) preserves the validity of hypothesis test statistics from *de novo* estimation in the holdout confirmatory sample with 3) all findings publicly documented in an earlier registered report and in this journal publication. Bayesian estimation procedures that leverage information from the exploratory stage included in the confirmatory stage estimation replace traditional frequentist null hypothesis significance testing. In addition to increasing statistical power by using information from the full sample, Bayesian methods directly estimate a probability distribution for the magnitude of an effect, allowing much richer inference. Estimated magnitudes of diversity along academic discipline, race, ethnicity, and foreign-born status dimensions are positively associated with innovation. A maximally diverse ownership team on these dimensions would be roughly six times more likely to report new-to-market innovation than a homophilic team.

## Introduction

The credibility crisis in applied research extends far beyond science, diminishing the quality of public debate on contentious issues such as climate change, public health, and the value or triviality of promoting diversity in a liberal democracy. The admonition to "trust science" rings hollow when findings are not reproducible, fail to replicate in similar studies, and are not fully transparent concerning their derivation or testing of alternative hypotheses [1]. Increasing transparency and replicability is something that applied researchers can proactively incorporate in their research designs.

to researchers with an approved proposal through the Standard Application Process at a Federal Statistical Research Data Center (https://manager.researchdatagov.org/RDG_User_Guide.pdf). U.S. citizenship is not required for access to confidential Title 13 and Title 26 data, but the researcher must have resided in the U.S. in the 3 years prior to submission. All statistical output from analyses of these data must be cleared by a Disclosure Avoidance Officer and/or Disclosure Review Board before public release.

**Funding:** Oak Ridge Associated Universities (ORAU) under DOE contract number DE-SC0014664 (TRW) Willard R. Sparks Chair in Agricultural Sciences & Natural Resources (DML).

**Competing interests:** he authors have declared that no competing interests exist.

This article aims to provide proof of concept for a split sample/dual method research design applied to a highly contentious topic: the connection between the diversity of ownership teams and business innovation. The protocol corrects for two significant contributors to the fragility of findings: 1) by splitting the dataset into an exploratory sample for specification testing and a confirmatory sample for hypothesis testing, the validity of statistics that assume *de novo* tests are preserved; and 2) the validity of comparing numerous alternatives is preserved through false discovery rate and family-wise error rate corrections [2]. In addition, we significantly reduce the researcher's degrees of freedom in the selection of a diversity measure by requiring that the selected index satisfies four axioms pertinent to the analysis of diversity in small teams.

Our protocol's novel contribution replaces frequentist methods with Bayesian methods in the confirmatory stage of the analysis. This modification leverages information learned during the exploratory stage and uses these estimates as weakly informative priors in the confirmatory phase. The discrete steps of the protocol are outlined in S1 File. In addition to narrowing the credible interval of estimates, the method also increases the informational value of the estimates for inference. Frequentist and Bayesian confirmatory findings are presented together to assess the potential advantages of methodological pluralism.

The proof-of-concept also demonstrates an implementable solution to the problem of data-dependent analysis that plagues research that uses observational or secondary data [3]. The problem of data-dependent analysis, where unreported repeated specification testing produces fragile, often unreplicable findings, has long been recognized [4]. Preregistration has since been adopted widely for randomized control trials, but "it is unclear how to apply preregistration to the analyses of existing data, which account for the vast majority of social science. Development of practices appropriate for existing data—whether historical or contemporary, quantitative, or qualitative—is a priority" [5]. There have been a few proposed solutions, but there is no institutional mechanism for implementation [6, 7]. This problem could be corrected by the growing number of researchers using confidential data within the Census Bureau's Federal Statistical Research Data Centers (FSRDCs). Access to data is strictly controlled, so it would be possible to release confirmatory samples to researchers only after the publication of a pre-analysis plan, along with the complete set of findings estimated from an exploratory sample used to arrive at the plan. The use of confidential data is becoming common in economics. In 2022, 44 of 94 articles published in the *American Economic Review* were exempted from data sharing due to confidentiality restrictions [8].

## Diversity findings vulnerable to false discovery

The empirical problem that demonstrates the proposed protocol assesses how diversity within ownership teams is associated with a higher or lower probability of reporting new-to-market innovation. The data used are from the 2018 Annual Business Survey (ABS) produced jointly by the United States (U.S.) Census Bureau and the National Center for Science and Engineering Statistics within the U.S. National Science Foundation. The ABS replaced the Survey of Business Owners for employer firms, added the innovation module from the former Business R&D and Innovation Survey, and an R&D module for microbusinesses with fewer than ten employees. Demographic and background information for up to four owners for each firm is collected.

The hypothesis that diversity in ownership teams increases the incidence of innovation derives from a combinatorial conception of innovation: the bringing together of seemingly unrelated ideas [9]. For example, business owners coming from different academic specializations can be expected to combine ideas from different domains. Less clear is whether business

owners coming from very different life experiences mediated by, for example, race or country of birth might also generate more novel ideas [10]. Evidence that places characterized by greater diversity are also more innovative is the dominant paradigm in the literature compared to the counterclaim that regional homophily induces innovation [11–13]. A highly cited work by Putnam that provides evidence that diversity is negatively associated with social capital, trust, and altruism at the regional level recognizes the overwhelming evidence of a positive association with creativity and innovation [14].

Findings at the firm level of whether diversity or homophily is more likely to promote innovation are mixed. The hypothesis that diversity hinders innovation derives from the role that homophily may play in facilitating the flow of information [15, 16]. The two countervailing effects of diversity on the flow of information include affective conflicts—where the attitudes or emotions of one group are incompatible with others—and cognitive conflicts—where a divergence of ideas owing to different experiences may resolve as performance-enhancing synthesis [17].

Cognitive conflicts are the basis of the seminal theoretical explanation of the advantage of diversity over homophily in problem-solving [18]. Empirical strategies to better isolate cognitive conflicts from affective conflicts include the development of a measure of 'unusualness' that compares the nationalities making up firms to the cultural diversity of a region to capture differences in cognitive approaches [17]. Management studies have identified the benefits of homophily in start-ups or research partnerships regarding innovation [16, 19] and firm survival [19]. Given the potential for both affective and cognitive conflict to influence the effect of diversity on innovation, the direction of influence remains an empirical question. The possible confounding factor of discrimination that might thwart successful innovation from diverse ownership teams is neutralized in the present analysis by the positive measure of innovation available in ABS that is used in this study: novelty and substantive change from what was done before are the requisite criteria.

In addition to being a highly contentious topic with implications for decisions on diversity training, affirmative action, and immigration policy, the topic is highly vulnerable to false discovery. Numerous ways exist to measure diversity. Nijkamp and Poot [20] consider more than 20 such measures. Measures typically combine relevant attributes such as race, ethnicity, gender, or foreign-born status to proxy diversity. The proliferation of these instruments provides a research environment ripe for cherry-picking and prone to false discovery and a multiple comparison problem.

The first step in reducing the set of estimates is to select a single measure that satisfies the requisite axioms discussed below [21]. A single measure reduces the dimensionality of the multiple comparison problem. It eliminates inductive searches for measures that reinforce the researchers' priors. While the axiomatic approach does not necessarily guarantee an optimal measure, it does make the selection process transparent.

The axioms the diversity measure is required to satisfy include:

1. HOMOPHILY AXIOM: All owners belonging to the same group must result in the lowest diversity measure value.

2. FRACTIONALIZATION AXIOM: Increasing the number of groups must increase the diversity measure value.

3. TEAM SIZE AXIOM: Larger ownership teams not demonstrating homophily must increase the diversity measure value relative to smaller ownership teams.

4. CONCENTRATION OF OWNERSHIP AXIOM: Ownership concentrated in one team member must reduce the diversity measure value relative to ownership that is more equally distributed among team members.

The homophily axiom requires that the value of the diversity measure be the lowest for ownership teams that lack diversity in the dimension of interest. This feature raises the issue of how to deal with single-owner firms for which diversity along any dimension or across many dimensions is impossible. The approach used in the exploratory analysis that is followed here is to exclude single-owner firms from the analysis. This exclusion allows for a comparison of the association of homophilic and heterophilic collaboration with innovation that is not confounded by the effect of no-owner collaboration on innovation.

The fractionalization axiom requires the diversity measure to increase with the number of unique groups in any ownership team dimension. The seven dimensions used to analyze the association between diversity and innovation are age, educational level, ethnicity, sex, education specialization, race, and foreign-born status. The maximum number of unique groups in any dimension is four, corresponding to each owner in a four-owner firm belonging to a different group. Ownership team size limits the number of unique groups in any dimension. However, two unique groups characterize dimensions as binary in the 2018 ABS, namely sex or foreign-born status. Details on the unique groups in each dimension are provided in Table 1.

The team size axiom conflicts with the assumption that diversity is a function of relative composition defined by population shares, not dependent on population size. However, the argument can be made that this is where the construct of diversity as a population characteristic breaks down for understanding diversity in much smaller ownership teams. Conventional diversity measures indicate the likelihood of interacting with members of another group governed by the relative size of group shares. Within a small ownership team, the probability of interaction is 1. The issue is whether guaranteed interaction with more members of another group should result in a higher measure of diversity than guaranteed interaction with fewer members. Suppose an ownership team has members from two groups. In that case, the level of diversity from a conventional measure is the same, no matter the number of owners. Compare this with ownership teams of different sizes. In a two-person ownership team, each owner is exposed to a single alternative viewpoint from another group. In a 4-owner firm, each will be exposed to 2 viewpoints from another group. If the ownership team is conceived as a network, then the number of diverse nodes increases with the size of the network for all heterophilic teams.

The other salient difference between a population and an ownership team is the separation of the ownership interest from the characteristics of any individual owner. In contrast to "one person, one vote," any owner's relative power or influence may be dominant or nominal. The concentration of ownership axiom recognizes that considering diverse viewpoints is likely greater in a firm characterized by equal shares of ownership relative to the firm where one owner controls a dominant interest. Ownership shares can be treated as population shares found in more traditional diversity measures.

The commonly used ethnolinguistic fractionalization (E.F.) measure is invariant to team (population) size by design, so it fails as a candidate measure. However, two simple modifications of the E.F. result in a measure satisfying all four axioms. An ownership fractionalization measure is defined as:

$$OF = 1 - \sum_{i=1}^{o} p_i^n \tag{1}$$

where $p$ = ownership share, raised to the power of $n$ = the number of unique groups, summed over $o$ = the number of owners, produces a number between 0 (homophily) and near 1 (maximally diverse) for diversity on a single dimension.

**Table 1. Unique groups per diversity dimension and group descriptions.**

| Dimension | Unique Groups | Group Descriptions |
|---|---|---|
| Age | 6 | Under 25 |
| | | 25–34 |
| | | 35–44 |
| | | 45–54 |
| | | 55–64 |
| | | 65 or over |
| Educational level | 9 | Less than high school |
| | | High school graduate |
| | | Technical or trade school |
| | | Some college |
| | | Associate degree |
| | | Bachelor's degree |
| | | Master's degree |
| | | Doctorate degree |
| | | Professional degree |
| Ethnicity | 5 | Not Hispanic |
| | | Mexican |
| | | Puerto Rican |
| | | Cuba |
| | | Other Hispanic |
| Sex | 2 | Male |
| | | Female |
| Education specialization | 17 | Biological/agriculture/environmental life sciences |
| | | Chemistry, except biochemistry |
| | | Computer/mathematical/technology sciences |
| | | Earth/atmospheric/ocean sciences |
| | | Economics/political/psychology/sociology and other social sciences |
| | | Engineering |
| | | Health |
| | | Physics/Astronomy |
| | | Science/mathematics teacher education |
| | | Other science/engineering related fields, not listed above |
| | | Art and humanities fields |
| | | Education other than science/mathematics |
| | | Management and administration fields |
| | | Sales and marketing fields |
| | | Social service and related fields |
| | | Other non-science/non-engineering related fields not listed above |
| | | No 4-year degree or higher |
| Race | 14 | White |
| | | Black or African American |
| | | American Indian or Alaska Native |
| | | Asian Indian |
| | | Chinese |
| | | Filipino |
| | | Japanese |
| | | Korean |
| | | Vietnamese |
| | | Other Asian |
| | | Native Hawaiian |
| | | Guamanian or Chamorro |
| | | Samoan |
| | | Other Pacific Islander |

*(Continued)*

**Table 1.** (Continued)

| Dimension | Unique Groups | Group Descriptions |
|---|---|---|
| Foreign-born Status | 2 | Born in the U.S. |
| | | Not born in the U.S. |

Source: 2018 Annual Business Survey

We show that the proposed OF measure of diversity satisfies the four axioms: 1) it produces '0' in cases of complete homophily (one group); it increases as the number of groups increases (fractionalization); 3) it increases with larger ownership teams (team size); and 4) it decreases when ownership is concentrated in one member (concentrated ownership).

Consider first the axiom of *homophily*; all owners belonging to the same group must result in the lowest diversity measure value. If all owners are from the same group, then $n = 1$ (homophily), and the diversity measure should be zero. In this case, the owner's ownership share, $p_i$, will sum to one. Thus, $\sum_{i=1}^{o} p_i^1 = \sum_{i=1}^{o} p_i = 1$ when $n = 1$. Substituting this into the OF formula, $OF = 1 - \sum_{i=1}^{o} p_i = 1 - 1 = 0$ (e.g., no diversity). Therefore, the OF measure is zero when all owners are from the same group and is consistent with the lowest possible diversity, satisfying the homophily axiom.

Next, consider the second axiom of *fractionalization*. This axiom states that the diversity value of OF increases when the number of groups increases. When there are more groups, $n$ is larger. Since each ownership share is raised to the power of $n$, and $0 \leq p_i \leq 1$, then $p_i^n$ decreases. Therefore, $\sum_{i=1}^{o} p_i^n$ decreases as the number of groups increases, which leads to a larger value of OF, which satisfies the axiom of fractionalization.

Consider now the *team size* axiom. This axiom states that larger ownership teams that do not demonstrate homophily will increase the diversity measure relative to smaller ownership teams. If ownership is distributed among more owners (larger $o$) but ownership is not concentrated in a single group (i.e., no homophily), OF will increase. As set $o$ increases, while keeping the ownership shares $p_i$ distributed among them, $\sum_{i=1}^{o} p_i^n$ gets smaller. This result happens because each $p_i$ is an ownership fraction. With more owners, the $p_i^n$ will be smaller, on average. As a result, OF increases when there are more ownership teams, which satisfies the team size axiom.

Lastly, consider the *ownership* axiom. This axiom states that ownership concentrated in one team member must decrease the diversity measure value relative to ownership that is more equally distributed among team members. In other words, OF will decrease if one team member holds most or all of the ownership, which indicates lower diversity. When one owner holds all or a majority of ownership, the ownership share $p_1$ is close to one and the others' ownership shares are close to zero. This means $\sum_{i=1}^{o} p_i^n$ will be near one because $p_1^n \rightarrow 1$ and the others' ownership shares will be close to zero. Therefore, the OF gets smaller when ownership is concentrated.

Tables 2 and 3 compare the maximal values of the OF and E.F. indices for different ownership team sizes and number of unique groups. For both indices, the tables demonstrate

**Table 2. Maximal values of of index for different number of owners and number of unique groups.**

| | Owners: | | |
|---|---|---|---|
| **Unique Groups** | **2** | **3** | **4** |
| **1** | 0 | 0 | 0 |
| **2** | 0.5 | 0.666667 | 0.75 |
| **3** | | 0.888889 | 0.9375 |
| **4** | | | 0.984375 |

**Table 3. Maximal values of E.F. index for different number of owners and number of unique groups.**

|  | Owners: |  |  |
| --- | --- | --- | --- |
| Unique Groups | 2 | 3 | 4 |
| 1 | 0 | 0 | 0 |
| 2 | 0.5 | 0.5 | 0.5 |
| 3 |  | 0.666667 | 0.666667 |
| 4 |  |  | 0.75 |

satisfaction of the homophily and fractionalization axioms. The E.F. index violates the team size axiom that is satisfied using the OF index. In addition to satisfying all four axioms, the more extensive range of the OF index will result in more considerable variance across observations, a desirable characteristic for an independent variable tasked with explaining variance in a dependent variable. The strong correlation between the E.F and OF indices in the data indicates that the two measures are picking up similar phenomena with respect to diversity.

Finding an index that satisfies the required axioms for analyzing the potential impact of ownership diversity on innovation resolves the most severe problem of false discovery resulting from multiple index searches as part of specification searches. However, the number of dimensions of interest concerning diversity and innovation is significantly greater than one, implying that the multiple comparison problem has not been fully resolved. Consideration of composite diversity measures that combine diversity from multiple dimensions dramatically increases the possible number of comparisons. These seven dimensions result in 120 unique combinations. However, a composite ownership fractionalization measure that combines two or more diversity measures provides an empirical analog to the multidimensional character of diversity experienced in everyday life. An unweighted composite ownership fractionalization (COF) index is:

$$COF = \frac{(D - \sum_{i=1}^{D} \sum_{j=1}^{o} P_{ij}^{ni})}{D} \tag{2}$$

where $D$ = the number of diversity dimensions. Normalizing by $D$ ensures that estimates and odds ratios are comparable across measures. A weighted COF is possible where selected dimensions such as foreign-born status or race and education specialization are emphasized in analyses related to immigration or affirmative action policy, respectively. However, the unweighted COF index used here provides a fundamental baseline.

The 127 different single and composite diversity measures provide a concrete example of the multiple comparison problem. The hypothesis test statistic or $p$-value is the probability that the sample was drawn from the null distribution corresponding to no effect. If the null is true and diversity has no association with innovation in the population, then a nominal $p$-value of 0.05 or 0.01—i.e., the probability that the significant result is due to chance in 1 in 20 or 1 in 100 draws—for one or a handful of measures would be a meaningless hypothesis test statistic. A transparent protocol to avoid such false discoveries is described next.

## Split sample component

The split-sample half of the protocol follows Anderson and Magruder [2], who examine various trade-offs in the design of split-sample studies and their statistical power, thresholds for passing exploratory hypotheses on for confirmation, and the use of one-sided tests. Monte Carlo methods are used to arrive at a 35%/65% split between exploratory and confirmatory samples to maximize power in the confirmatory sample. This guidance is followed in the present analysis.

Sample size in the present case is less of an issue than the large reductions in power associated with false discovery rate and family-wise error rate corrections, discussed below.

The large sample size of the 2018 ABS (approximately 142,000 firms with multiple owners) does raise concerns regarding the most productive thresholds for passing exploratory hypotheses on for confirmation. Large samples have contributed to an inference crisis in economics where statistical significance is often uncritically equated with economic significance [22]. As the sample size increases, the likelihood of an estimate landing very close to zero, which is also statistically different from zero, increases. A pass-through threshold for magnitude is also applied to ensure that exploratory hypotheses passed through for confirmation are economically significant. Parameter estimates are required to meet a minimum effect size corresponding to a "small effect" and the conventional $p$-value $< 0.05$ threshold for statistical significance. The magnitude criterion used is an odds ratio of $\geq 1.44$ (or $< 0.6945$), corresponding to Cohen's $d$ of $\geq 0.2$ [23].

Finally, previous research finds that diversity and homophily have been positively associated with innovation. Two-sided tests are used for this reason [10, 16, 17, 19].

## Specification of the innovation and ownership diversity regression equation

Isolating the association between ownership diversity and business innovation requires controlling for potentially confounding variables. Alternative sources of variation that may be correlated with both diversity and innovation include the industry the business operates in and firm size. An empirical regularity in the ABS and other innovation surveys finds a higher incidence of innovation in larger firms [24, 25]. Larger firms are also more likely to have larger ownership teams, opening the possibility of greater diversity. Nine firm size categories are included in the regression to control for this source of variation: micro companies (1–4 and 5–9 employees), small companies (10–19, 20–49), medium companies (50–99, 100–249), and large companies (250–499, 500–999, 1000 or more).

The possible interaction between industry, innovation, and diversity is most salient for family-owned businesses. Construction and Accommodation and Food Services are two industries that may have relatively lower levels of innovation but may be more homophilic concerning race and ethnicity and potentially more diverse for sex and age owing to the prevalence of family ownership. Two-digit North American Industry Classification System (NAICS) controls for all industries other than Agricultural Services (07) and Public Administration (91–92) are included in the statistical model to provide better estimates of the independent effect of diversity on innovation. Other controls could be included, but industry and firm size are noncontroversial and provide a parsimonious specification:

$$\Pr(y = 1) = \text{logit}(\beta_0 + \beta_1 \cdot x_1 + \mathbf{FE}_{firmsize}\mathbf{A}_1 + \mathbf{FE}_{industry}\mathbf{A}_2) \qquad (3)$$

where:

$y$ = self-reported new-to-market innovation (binary variable);
$x_1$ = OF or COF index;
$\mathbf{FE}_{firmsize}$ = firm size fixed effects array (with parameter $\mathbf{A}_1$);
$\mathbf{FE}_{industry}$ = industry fixed effects array (with parameter $\mathbf{A}_2$);
logit = the logistic link function.

The dependent variable "$y$" is a self-report indicating the introduction of a novel or significantly improved product that was "new to the market" in the preceding three years. Responses to this innovation question were more likely to represent more far-ranging, novel innovation and less likely to include incremental innovation than "new to the business" or "process"

innovation questions [26]. The response to the new-to-market innovation question is thus the most appropriate for assessing how diversity may spawn or inhibit more novel ideas.

## Exploratory results

The complete set of results from the 127 logistic regressions is included in a published Registered Report [27]. The diversity measures passed through are included as priors in the last column of S2 File. Only ten equations contained a diversity measure that failed to meet significance and magnitude thresholds (S3 File). All the diversity measures passed through for confirmation were positively associated with new-to-market innovation. All the measures not passed through for confirmatory testing included sex and/or age—the two dimensions most likely to demonstrate considerable diversity in family-owned businesses. Owners carrying on a family business had the lowest incidence of new-to-market innovation relative to all other reasons for owning a business in the 2018 ABS [24]. Future research examining business ownership diversity should consider controlling for family-owned firms or firms jointly owned by a couple.

While the commonsense conclusion that the vast number of measures passed through for confirmation should reinforce our belief in all forms of diversity increasing innovation, the implication from frequentist inference is the opposite. The multiple comparison problem from 117 measures imposes a very strict threshold $p$-values in correcting for a family-wise error rate (FWER). The threshold required to ensure an error rate of 0.05 for the complete set of estimates is, using Bonferroni's method, 0.05/117, or a $p$-value $< 0.000427$. The less conservative false discovery rate (FDR) requires an average error rate of 0.05 for all estimates ($m$) and is calculated by ranking the estimates from most ($i = 1$) to least precise ($i = m$) and using 0.05 x ($i/m$) as the correction. Intuitively, the most precise estimate requires the FWER correction, and the least precise estimate requires the nominal 0.05 threshold to be deemed statistically significant.

The drastic reduction in statistical power as the $p$-value decreases results from the sensitivity of frequentist hypothesis tests to the number of unique looks at the data. However, within the split sample protocol, frequentist methods impose a second considerable reduction in power relative to the complete sample. Because there is no way to incorporate prior information in frequentist estimation, none of the information from the exploratory sample is available for the confirmatory tests. The sample size has been reduced by 35%, reducing power. Bayesian methods can incorporate prior information during model estimation and thus potentially regain some of the power otherwise lost in a full frequentist split sample protocol.

## Dual method component

The pragmatic appeal of using Bayesian methods to preserve statistical power in a split sample protocol runs against the widespread philosophical aversion to these methods in applied research [28–31]. However, the cost the researcher is willing to pay in reduced power to improve transparency by splitting the sample can pay dividends in mollifying empiricist objections. Most importantly, the fundamental question of the possible existence of an effect that the frequentist approach excels at is answered in the exploratory stage [22]. This benefit extends to calibration and specification testing advantages, which are also accomplished in the exploratory stage [32–34].

The benefits of adopting Bayesian estimation in the confirmatory stage include 1) leveraging information learned from the full sample by using the exploratory point estimates that pass through as weakly informative priors, 2) replacing the indirect frequentist estimation of Pr (sample data | $H_0$) with the direct estimation of Pr(phenomenon | sample data), that results in

3) the ability to inform decision-making even when prior information updated with new evidence suggests the phenomenon has little or no effect.

The philosophical aversion to Bayesian applications is reinforced by unfamiliarity with the procedures in statistical software that tend to require substantially more user input than frequentist procedures [32]. The exploratory stage provides some of this input but not enough to proceed to estimation. What we used from the exploratory stage and how we arrived at reasonable values for the missing input should assist those with little experience with Bayesian estimation.

The priors used to estimate the model parameters in Eq 1 are weakly informative. The priors for the intercept, $\beta_0$ is the normal distribution centered on zero with a standard deviation of 10. The standard deviation of 10 corresponds with a prior variance of 100. Setting the variance to this value means the prior distribution is centered over the parameter value from the exploratory findings but with very wide, symmetric tails. Bayesian procedures use simulation methods to generate posterior distribution conditional on prior information. The wide tails associated with a standard deviation of 10 ensure a more extensive search space around the prior learned from the exploratory step, allowing the sampling procedure to explore the posterior distribution more fully.

The priors for the parameters on firm size and the NAICS industry classification are also normal but centered on zero, with a standard deviation of 10. The priors for OF and COF are centered on the estimates for these variables from the exploratory results (Tables 4 and 5), with a standard deviation of 10. The Bayesian estimates' highest posterior density intervals (hdpi) are reported as confidence intervals. Intervals that include zero indicate that the estimate is not different from zero. The tail probabilities were adjusted using the Bonferroni Type I error rate of 0.0004273.

The Bayesian estimation procedures must also specify diagnostics for assessing convergence of the Monte Carlo Markov Chain (MCMC) simulation. The most accessible, intuitive diagnostic tool is a trace plot that plots the parameter value against the iteration number [35]. Convergence and stationarity can be qualitatively assessed with traces that fluctuate rapidly around

**Table 4. Comparing most precise frequentist and bayesian log odds estimates of ownership fractionalization on new-to-market innovation.**

| Ownership Fractionalization | ————————Frequentist———————— | | | | | | | ————————Bayesian———————— | | | | | |
|---|---|---|---|---|---|---|---|---|---|---|---|---|---|
| | Log Odds | Std Err | p-value | FDR α = 0.05 Threshold | Odds Ratio | Bonferroni Lower CI Odds Ratio | Bonferroni Upper CI Odds Ratio | Equal Tailed Lower (alpha = 0.000427) | Equal Tailed Upper (alpha = 0.000427) | Mean Odds Ratio | Mean Log Odds | Std Dev | Exploratory Estimate & Bayesian Prior |
| MRU[g] | 1.586 | 0.0953 | 0.000000013 | 0.000427 | 4.887 | **2.989** | **7.989** | **4.214** | **5.789** | 4.855 | 1.58 | 0.0517 | 1.588 |
| HMRU[g] | 1.884 | 0.1135 | 0.000000013 | 0.000855 | 6.580 | **3.665** | **11.81** | **5.157** | **10.09** | 6.828 | 1.921 | 0.1043 | 1.955 |
| HMR[g] | 1.760 | 0.1246 | 0.000000062 | 0.001282 | 5.815 | **3.059** | **11.05** | **4.957** | **7.532** | 6.013 | 1.794 | 0.0702 | 1.799 |
| EHMRU[ghw] | 1.720 | 0.1223 | 0.000000065 | 0.001709 | 5.587 | **2.973** | **10.49** | **4.354** | **7.412** | 5.703 | 1.741 | 0.0949 | 1.978 |
| MU[ghw] | 1.200 | 0.0858 | 0.000000068 | 0.002137 | 3.321 | **2.134** | **5.169** | **2.854** | **3.702** | 3.267 | 1.184 | 0.0425 | 1.212 |
| HMU[g] | 1.571 | 0.1129 | 0.000000072 | 0.002564 | 4.814 | **2.689** | **8.616** | **3.596** | **6.187** | 4.697 | 1.547 | 0.0875 | 1.653 |
| EMRU[ghw] | 1.471 | 0.1058 | 0.000000072 | 0.002991 | 4.354 | **2.524** | **7.512** | **3.542** | **5.25** | 4.297 | 1.458 | 0.0664 | 1.654 |
| EHMR[ghw] | 1.541 | 0.1113 | 0.000000076 | 0.003419 | 4.668 | **2.629** | **8.289** | **3.794** | **5.916** | 4.655 | 1.538 | 0.0766 | 1.777 |
| EMR[g] | 1.228 | 0.0917 | 0.000000104 | 0.005385 | 3.415 | **2.128** | **5.480** | **2.792** | **4.176** | 3.438 | 1.235 | 0.0629 | 1.375 |
| MR | 1.331 | 0.1019 | 0.000000131 | 0.004273 | 3.787 | **2.239** | **6.407** | **3.192** | **4.63** | 3.834 | 1.344 | 0.0604 | 1.291 |

*Source*: *2018 Annual Business Survey 65% Test Sample, project number P-7504866, Disclosure Review Board approval number CBDRB-FY23-0335. FSRDC project number 2681. Notes*: *A = Age, E = Educational Level, G = Sex, H = Ethnicity, M = Education Specialization, R = Race, and U = Foreign-born Status.* [g] indicates posterior distribution failed to pass Geweke's test for stationarity, [hw] indicates posterior distribution failed to pass Heidelberg-Welch convergence diagnostic. Industry and firm size fixed effects are not shown.

**Table 5. Comparing least precise frequentist and bayesian log odds estimates of ownership fractionalization on new-to-market innovation.**

| Ownership Fractionalization | —Frequentist— | | | | | | | —Bayesian— | | | | | |
|---|---|---|---|---|---|---|---|---|---|---|---|---|---|
| | Log Odds | Std Err | p-value | FDR α = 0.05 Threshold | Odds Ratio | Bonferonni Lower CI Odds Ratio | Bonferroni Upper CI Odds Ratio | Equal Tailed Lower (alpha = 0.000427) | Equal Tailed Upper (alpha = 0.000427) | Mean Odds Ratio | Mean Log Odds | Std Dev | Exploratory Estimate & Bayesian Prior |
| AU[ghw] | 0.3057 | 0.0823 | 0.004 | 0.04615 | 1.358 | **0.888** | **2.076** | **1.170** | **1.589** | 1.344 | 0.2959 | 0.0491 | 0.4043 |
| AEGH[g] | 0.3129 | 0.0946 | 0.0079 | 0.04658 | 1.367 | **0.839** | **2.227** | **1.135** | **1.604** | 1.359 | 0.3068 | 0.0554 | 0.5859 |
| EG[ghw] | 0.2148 | 0.0659 | 0.0086 | 0.04701 | 1.24 | **0.883** | **1.741** | **1.094** | **1.388** | 1.244 | 0.2186 | 0.0374 | 0.3928 |
| AEG[hw] | 0.2293 | 0.0764 | 0.0133 | 0.04743 | 1.258 | **0.848** | **1.865** | **1.071** | **1.417** | 1.247 | 0.2211 | 0.0491 | 0.4163 |
| H[hw] | 0.1803 | 0.0667 | 0.0221 | 0.04786 | 1.198 | **0.849** | **1.689** | **0.921** | **1.499** | 1.215 | 0.1945 | 0.0918 | 0.4541 |
| GHU[hw] | 0.2941 | 0.1085 | 0.0219 | 0.04829 | 1.342 | **0.813** | **2.216** | **1.020** | **1.624** | 1.312 | 0.2715 | 0.0809 | 0.5313 |
| AGHU[g] | 0.3063 | 0.1166 | 0.0253 | 0.04872 | 1.358 | **0.744** | **2.478** | **1.031** | **1.69** | 1.314 | 0.2733 | 0.0744 | 0.5522 |
| AGHR | 0.2576 | 0.1117 | 0.0438 | 0.04914 | 1.294 | **0.727** | **2.301** | **1.010** | **1.658** | 1.296 | 0.2592 | 0.0798 | 0.4876 |
| AGU[g] | 0.2234 | 0.0974 | 0.0447 | 0.04957 | 1.250 | **0.757** | **2.066** | **1.072** | **1.492** | 1.265 | 0.2347 | 0.0554 | 0.3802 |
| GHR[g] | 0.2081 | 0.1016 | 0.0678 | 0.05000 | 1.231 | **0.729** | **2.080** | **0.957** | **1.572** | 1.224 | 0.2022 | 0.0909 | 0.4183 |

Source: 2018 Annual Business Survey 65% Test Sample, project number P-7504866, Disclosure Review Board approval number CBDRB-FY23-0335. FSRDC project number 2681.

*Notes*: A = Age, E = Educational Level, G = Sex, H = Ethnicity, M = Education Specialization, R = Race, and U = Foreign-born Status. [g] indicates posterior distribution failed to pass Geweke's test for stationarity, [hw] indicates posterior distribution failed to pass Heidelberg-Welch convergence diagnostic. Industry and firm size fixed effects are not shown.

the mode—described as a furry caterpillar—indicating a more reliable posterior distribution. In contrast, wandering parameter values indicate potential problems with the sample, model, or priors.

Another qualitative tool is the Bayes effective sample size (ESS), with values above 400 generally regarded as a minimum for reliable simulations but preferably above 1,000 [36]. Diagnostic statistics providing tests of stationarity and convergence include Raftery-Lewis, Gelman and Rubin, Geweke [37], and Heidelberger-Welch [38]. The latter two are used in this analysis. They provide information concerning the Markov chain's first (mean) and second (variance) moments. The Geweke procedure tests if the mean estimates of a chain have converged (i.e., they are stationary) by comparing the means from the early and later segments of the posterior sample. Using a different statistical procedure, the Heidelberger-Welch (H.W.) tests whether the covariance of the posterior distribution is stationary.

A unique addition for this demonstration is using the FWER threshold *p*-value to construct the equal-tailed credible interval so that it is comparable to the corrected frequentist confidence interval. In addition to allowing a comparison of statistical power between the two methods, a credible interval can also deliver the familiar verdict of "statistically different from zero" or "suspend judgment."

## Confirmatory results

The complete results for all 117 diversity measures passed through for confirmatory testing are in S2 and S4 Files. The discussion of findings is limited to inferences from the extremes of the diversity measure estimates: the most precise estimates tend to have the largest magnitudes, and the least precise estimates tend to have the smallest magnitudes. Replacing the OF diversity measures with E.F. diversity measures produced qualitatively similar results but of smaller magnitude (not reported).

Table 4 includes the *de novo* results of the ten most precise diversity measure estimates using the confirmatory sample. Exponentiating the log odds estimates results in odds ratios

that are interpreted as the change in the odds of an event occurring (report of new-to-market innovation), given a one-unit change in the independent variable. The most considerable magnitude is for the diversity measure, including Ethnicity, Education Specialization, Race, and Foreign-born Status, where a maximally diverse ownership team (COF = 0.8672) would be roughly six times as likely to report new-to-market innovation relative to an ownership team that was homophilic on these dimensions. The most significant difference between the frequentist and Bayesian estimates is the confidence interval width and credible interval, respectively. The frequentist confidence interval that includes no prior information on the magnitude of the coefficient estimate ranges from an increase in odds from 3.2 to 10.2, whereas the comparable Bayesian credible interval is an increase in odds between 4.5 and 8.75. The narrowing of the confidence is unsurprising. Bayesian estimators belong to a broader class of James-Stein shrinkage estimators [39].

The family-wise error rate and false discovery rate corrections are in agreement concerning statistical significance in Table 4, which reports the most precise estimates. This agreement is not evident in Table 5, which includes *de novo* results from the least precise estimates. All but one of the estimates is statistically significant using the false discovery rate correction. However, none are statistically significant using the FWER correction (i.e., odds ratios that are less than and greater than one within the confidence interval). This divergence demonstrates the increase in power from the Bayesian estimation. Despite using the same stringent FWER corrected *p*-value in constructing the equal-tailed credible interval, the Bayesian estimate delivers the same verdict as the much less conservative FDR correction. In fact, 28 of the 117 (24%) diversity measure estimates are insignificant after applying the FWER correction. The best antidote for increasing the replicability and credibility of applied research is increased statistical power [40]. Bayesian estimation in the confirmatory stage can help deliver it.

The results presented are from *de novo* estimations using the confirmatory sample, consistent with the split sample protocol [2]. The single run that ensures the frequentist test statistics are valid may be at odds with a Bayesian workflow needed to ensure reliable posterior distribution estimates. Eighteen of the twenty Bayesian parameter estimates in Tables 4 and 5 fail to pass convergence and/or stationarity tests provided by the Geweke and Heidelberger-Welch diagnostics. This lack of convergence is unsurprising given the large number of parameters in the model to control for industry and firm size class. The fixed effects for industry and firm size indicators effectively soak up unobserved confounds in the frequentist approach. However, when applied in the Bayesian estimation, these fixed effects result in a high degree of autocorrelation, resulting in poor mixing and no convergence (S5 File). Estimating the pooled model substantially reduces the autocorrelation, which results in much more rapid mixing, with all convergence and stationarity tests passed. However, these diversity estimates also tend to be inflated due to a failure to control for possible confounds.

A regular part of a Bayesian workflow is implementing various strategies to reduce autocorrelation and increase mixing so that estimates are derived from reliable posterior distributions. Various strategies include thinning to break up the autocorrelation between the draws, substantially increasing the number of iterations, or standardizing variables to decrease the posterior correlation between parameters. The approach followed here was to reconsider the appropriateness of fixed effects for industry and firm size. A correlated random effects specification—commonly referred to as the Mundlak approach [41]—was tried. However, the parameters failed to pass various diagnostic tests. A mixed-effects specification (S6 File) where industry effects were treated as fixed and firm size effects treated as random resulted in a rapid reduction in autocorrelation, much better mixing, and passing diagnostic tests with the effective sample size generally well above 1,000 for the diversity parameter in the estimations (S7 File). These results, provided in S4 File, are qualitatively very similar to the *de novo*

confirmatory estimates but represent final estimates of the effect of team diversity on innovation as the failures to pass the diagnostics test in the *de novo* estimation are resolved.

The rationales for presenting the *de novo* estimates, many of which failed diagnostic tests, in the body of the article are 1) to strictly follow the pre-analysis plan prescribed in the exploratory study, clearly demonstrating the steps required of the split sample/dual method protocol; 2) allow a direct comparison with the *de novo* frequentist estimates to demonstrate the potential advantages of methodological pluralism that are not obscured by differences in model specification; and 3) to normalize the additional steps in a Bayesian workflow that may be required to get valid estimates from a model first specified for frequentist estimation. Subsequent use of the split sample/dual method protocol that does not serve as proof of concept can reverse this presentation, putting the final results in the body of the article and *de novo* Bayesian estimates failing diagnostic tests in an appendix. However, transparency requires documenting any required specification change to derive valid Bayesian estimates.

All but one of the diversity measures estimated in Tables 4 and 5 are composite, multidimensional measures. This feature raises the question of the relative contribution of each diversity dimension to new-to-market innovation. The diversity dimensions most common in the most precisely estimated diversity measures that also have the largest magnitudes are reported in Table 4 are Education Specialization, Foreign-born Status, Race, and Ethnicity. In contrast, the diversity dimensions most common in the Table 5 diversity measures estimated with the least precision and tend to have the smallest magnitudes are Sex and Age. As noted earlier, these results should be interpreted with caution due to the confounding factor of family ownership that is not controlled. A regression decomposition that estimates the contribution of each included dimension to the log odds of the respective diversity measure estimate in Table 6 allows comparing the relative contribution of each diversity dimension. The same split sample/dual method approach was used for the diversity measure estimates, i.e., estimates from the frequentist analysis applied to the exploratory sample are used as weakly informative priors in the Bayesian estimation.

Education Specialization is the diversity dimension with the least controversial connection to innovation, and it also has the most considerable effect on the log-odds ratio in the measures where it is included. The contribution from disciplinary diversity is roughly twice that of Race, Foreign-born Status, or Ethnicity and nearly three times that of Educational Level. The decomposition provides strong evidence that the novel combination of ideas comes from different lived experiences and rational ways of understanding developed within academic disciplines.

**Table 6. Posterior summaries of bayesian regression decomposition of diversity attributes and new-to-market innovation.**

| Variable | Mean | Standard Deviation | 95% Highest Posterity Density Credible Interval | |
|---|---|---|---|---|
| Intercept | 0.2381 | 0.0214 | 0.1946 | 0.2726 |
| Age (A) | -0.1862 | 0.0171 | -0.2174 | -0.1533 |
| Educational Level (E) | 0.1626 | 0.0193 | 0.1321 | 0.2019 |
| Sex (G) | -0.1850 | 0.0200 | -0.2281 | -0.1507 |
| Ethnicity (H) | 0.2404 | 0.0174 | 0.2058 | 0.2684 |
| Education Specialization (M) | 0.5572 | 0.0161 | 0.5247 | 0.5828 |
| Race (R) | 0.2618 | 0.0167 | 0.2260 | 0.2898 |
| Foreign-Born (U) | 0.2519 | 0.0192 | 0.2117 | 0.2825 |
| $\hat{\sigma}^2$ | 0.0324 | 0.00419 | 0.0250 | 0.0416 |

Source: 2018 Annual Business Survey 65% Test Sample, project number P-7504866, Disclosure Review Board approval number CBDRB-FY23-0335. FSRDC project number 2681.

This finding should prompt debate and additional rigorous analysis. However, any analysis that is not fully transparent should be substantially discounted because a data-dependent analysis is prone to false discovery.

## Discussion

Peer review can be expected to flag egregious cases of *p*-hacking or data dredging [42]. However, data-dependent analysis has proved to be far more insidious. Journal editors and referees have been far less vigilant in vetting questionable research practices that contribute to false discovery among conscientious researchers: conducting hypothesis tests on the same data used for numerous—though unreported—specification tests, failure to apply multiple comparison corrections to test statistics, or HARKing (hypothesizing after results are known) [43–47]. Replication studies may provide some correction for false discoveries, but the record so far is not encouraging. In addition to the difficulty of publishing replications [48–50], evidence suggests that publications that fail to replicate receive more citations than studies with replicable findings and that replications are rarely cited even if they diagnose substantive problems [51–52].

Increasing research transparency at the front end would appear to be the most feasible strategy for increasing the credibility of research and its contribution to public debate [53]. In the absence of scientific misconduct, there are no mechanisms for retracting findings that misconstrue phenomena in the population due to cherry-picking results that reinforce a researcher's priors. The cost of a lack of transparency is borne disproportionately by contentious issues of urgent social concern, such as climate change, public health, and diversity [54, 55]. Fierce debates may remain regarding the best way to conceptualize and design studies and the valid way to interpret findings. However, debates with asymmetric information that characterize the current situation—where specification testing preceding hypothesis testing is rarely reported —are inefficient and potentially detrimental.

The split sample/dual method protocol demonstrated here resolves this central problem of asymmetric information plaguing applied research using NHST. All specification testing results are published in a Registered Report [27] that provides a pre-analysis plan for *de novo* estimation using the holdout confirmatory sample, producing the preregistered replication recommended by Gelman and Loken [3]. The cost in statistical power incurred by providing this greater transparency can be partly retrieved by replacing frequentist estimation with Bayesian estimation in the confirmatory stage, exploiting the information in exploratory estimates passed through as weakly informative priors. By replacing the estimation of $\Pr(\text{sample data} \mid H_0)$ with $\Pr(\text{phenomenon} \mid \text{sample data})$, the confirmed findings from the analysis also allow for much richer inference. Finally, the protocol could be implemented within the Federal Statistical Research Data Center (FSRDC) system, providing an institutional guarantee of the strict separation of the data used for specification testing from the data used for hypothesis testing. The one constraint imposed by FSRDCs that may prevent full transparency, requiring publication of findings from all specification tests from the exploratory stage, is the limit on statistical output that can be released publicly. Disclosure risk increases as more findings from the same sample are released, and very similar specification tests might amplify this risk [56]. While it is unlikely that the U.S. Census Disclosure Review Board would clear all quantitative output from an exploratory analysis, the sign and significance of coefficient estimates for specifications not passed through for confirmation should be releasable.

The emphasis of this article has been on proof of concept for a research design capable of increasing transparency on contentious issues that can be applied to observational data [5]. Applied to the contentious issue of the potential contribution of diversity to innovation, the

implications of these findings cannot be summarily discounted as being cherry-picked to reinforce our priors. The strong association between diversity in terms of race, ethnicity, foreign-born status, and education specialization raises important policy questions regarding immigration and affirmative action policy. However, a critical limitation of this analysis is that causation has not been established. For this reason, any discussion of policy implications from these findings would be premature. Two areas of follow-on research to be pursued include the development of research designs capable of controlling for endogeneity resulting from unobserved heterogeneity (e.g., arising from diversity available in the local population [17]) and extending the analysis from the positive self-reported innovation measures requiring only novelty to innovation measures that either include market success as a criterion or that are mediated by a third party (e.g., patents awarded, securing venture capital).

## Supporting information

**S1 File. Exploratory and confirmatory analysis steps.**
(DOCX)

**S2 File. De Novo frequentist and Bayesian estimates.**
(DOCX)

**S3 File. Diversity indices not passed through.**
(DOCX)

**S4 File. Final and De Novo Bayesian estimates.**
(DOCX)

**S5 File. Diagnostic plot using PROC QLIM.**
(DOCX)

**S6 File. Representative SAS code.**
(DOCX)

**S7 File. Diagnostic trace plot using PROC BGLIMM.**
(DOCX)

## Acknowledgments

The Census Bureau has reviewed this data product to ensure appropriate access, use, and disclosure avoidance protection of the confidential source data used to produce this product. This research was performed at a Federal Statistical Research Data Center under FSRDC Project Number 2681 and the National Center for Science and Engineering Statistics Survey Sponsor Data Center under Project Number P-7504866. (CBDRB-FY23-0335 & CBDRB-FY24-0126). All opinions, findings, conclusions, and recommendations expressed in this paper are those of the authors and do not reflect the views of the National Center for Science and Engineering Statistics, National Science Foundation, Department of Energy, Oak Ridge Affiliated Universities, Oak Institute for Science and Education, or the U.S. Census Bureau.

## Author Contributions

**Conceptualization:** Timothy R. Wojan, Dayton M. Lambert.

**Data curation:** Timothy R. Wojan.

**Formal analysis:** Timothy R. Wojan.

**Investigation:** Timothy R. Wojan.

**Methodology:** Timothy R. Wojan, Dayton M. Lambert.

**Validation:** Timothy R. Wojan, Dayton M. Lambert.

**Writing – original draft:** Timothy R. Wojan, Dayton M. Lambert.

**Writing – review & editing:** Timothy R. Wojan, Dayton M. Lambert.

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
