## [Decision Letter · Decision Letter 0]

10 Jun 2024

PONE-D-24-08785Research Transparency on Contentious Topics: Exploring the Connection between Diversity and InnovationPLOS ONE

Dear Dr. Wojan,

Thank you for submitting your manuscript to PLOS ONE. After careful consideration, we feel that it has merit but does not fully meet PLOS ONE’s publication criteria as it currently stands. Therefore, we invite you to submit a revised version of the manuscript that addresses the points raised during the review process.

 Please respond to all comments and highlight in the ms.

We look forward to receiving your revised manuscript.

Kind regards,

Thiago P. Fernandes, PhD

Academic Editor

PLOS ONE

Journal Requirements:

Reviewers' comments:

Reviewer's Responses to Questions

**Comments to the Author**

1. Is the manuscript technically sound, and do the data support the conclusions?

Reviewer #1: Partly

Reviewer #2: Yes

Reviewer #3: Partly

2. Has the statistical analysis been performed appropriately and rigorously? 

Reviewer #1: No

Reviewer #2: Yes

Reviewer #3: I Don't Know

3. Have the authors made all data underlying the findings in their manuscript fully available?

Reviewer #1: Yes

Reviewer #2: Yes

Reviewer #3: Yes

4. Is the manuscript presented in an intelligible fashion and written in standard English?

Reviewer #1: Yes

Reviewer #2: Yes

Reviewer #3: Yes

5. Review Comments to the Author

Reviewer #1: Research Transparency on Contentious Topics: Exploring the Connection between

Diversity and Innovation

The statistical framing of this article in the abstract does not seem problematic but it is not new or a contribution. Similarly, the discussion in the introduction about the credibility crisis in applied research is also unrelated to the contribution of this article. Following best practices (or well known practices) is a necessary condition of publication, but not sufficient and not the points to lead with.

I would remove both.

The contribution seems to be about the causal effect of diversity of ownership teams on business innovation. I suggest leading with that.

The paper claims that “The first step in reducing the set of estimates is to select a single measure that satisfies the requisite axioms discussed below“, but why is a single measure required? Why can’t we study multiple dimensions of this multidimensional concept? Having a single measure is convenient, but what’s the reason for this? If the separate dimensions (your axioms and other features) are correlated, you’ll only learn that if you study the dimension separately, at least at first.

If you do have one dimension, and we try to satisfy the (reasonable) axioms proposed, then the authors claim that they have a measure proposed satisfies them. This is merely stated. It seems true, but it needs to be proven. But more importantly, is this the unique measure that satisfies the axioms? A proof that this is unique would be useful. (A somewhat similar example of a uniqueness proof, done by building on entropy, is in doi:10.1093/pan/mpl011). If not, then do the empirical results vary depending on which measure (among those that fit the axioms) is used?

The 2018 *annual* business survey seems like a useful dataset, but why stop in 2018? The article has the laudable training/test framework, and so why not -- following that setup -- use a purely out of sample test set with one of the subsequent years of data?

A causal identification assumption is required to make [Disp-formula pone.0313826.e001] useful. This needs to be clearly stated and justified. I’m not sure that is possible here, but it at least needs to be addressed. It is the central point of the entire paper.

[Disp-formula pone.0313826.e001] also makes big homogeneity assumptions (such as beta1 not varying by the fixed effects). These should be tested and explored. I would suggest that the authors first run their analysis separately within different firm size and industry classes and then only put them together into one equation if the data support that. More likely there are different types of effects in different industries or types of firms, but this will be apparent if you look rather than assume.

The “registered report” framework here is hurting the analysis. The authors are not learning from the data and so the readers are not learning from the authors. I would jettison that report and just do a proper model-free statistical exploration.

Reviewer #2: The manuscript demonstrates a robust and technically sound methodology, utilizing both frequentist and Bayesian approaches to enhance the reliability of findings. The use of a split-sample/dual-method protocol is a novel approach that enhances research transparency and addresses issues of reproducibility.The study addresses a highly relevant and contentious topic with significant implications for diversity and innovation policies.However, there are still few comments and suggestions for improvement as follow:

(1) Complexity of Methodological Explanation: The explanation of the dual-method approach and the statistical techniques used, including Bayesian methods and corrections for multiple comparisons, is highly technical and may be challenging for readers without a strong statistical background. Therefore, it is advisable to Simplify the methodological sections by providing a more straightforward summary or an appendix with detailed explanations and step-by-step guides. Including visual aids such as diagrams to illustrate the research design and statistical methods could also be beneficial.

(2) More Discussion on Practical Implications is needed: The manuscript provides an extensive methodological discussion but lacks depth in discussing the practical implications of the findings for business practices and policy-making, authors are advised to explain further how findings could influence business strategies, diversity policies, and innovation practices.

(3) Proofreading Needs, a final proofread to catch minor typographical errors and ensure consistency in terminology and formatting would improve the manuscript's polish.

(4) Two Self-Citation were found, While self-citations are acceptable, ensuring they do not dominate the reference list

Reviewer #3: Dear authors,

this is a very interesting paper pursuing several ambitious and relevant goals. I absolutely agree with the need for increased research transparency in social sciences/economics and appreciate the authors’ engagement with these issues and the proposed ideas and protocol. However, considering the different aims and contributions made, the paper would benefit from a clearer focus. This includes, on the one hand, engaging a little more with the existing literature on innovative effects of diversity and, on the other hand, ensuring that the proposed protocol is accessible for the intended audience.

Major points

1. I understand that the conceptual and empirical background on diversity effects is not the focus of this paper and is therefore kept short. However, as an application addressing (in the authors’ words) a “contentious” issue, and certainly one attracting both theoretical, empirical and policy attention, I find the engagement with the literature superficial. I broadly agree with the theoretical reasoning for both innovative advantages and hinderances but would suggest to acknowledge the existing literature on diversity effects a bit more explicitly. This applies especially to the section “Diversity Findings Vulnerable to False Discovery” but also to the sections discussing the results. Some suggestions:

o for an overview of the literature: Ozgen, C. (2021). The economics of diversity: Innovation, productivity and the labour market. Journal of Economic Surveys, 35(4), 1168–1216. https://doi.org/10.1111/joes.12433

o for a specific application of team diversity to firm innovation: Brixy, U., Brunow, S., & D’Ambrosio, A. (2020). The unlikely encounter: Is ethnic diversity in start-ups associated with innovation? Research Policy, 49(4), 103950. https://doi.org/10.1016/j.respol.2020.103950

o as a seminal reference for the existence of diversity effects: Page, S. E. (2007). The difference: How the power of diversity creates better groups, firms, schools, and societies. Princeton Univ. Press.

o for a review of different measures of diversity: : Nijkamp, P., & Poot, J. (2015). Cultural diversity: A matter of measurement. In The Economics of Cultural Diversity (pp. 17–51). Edward Elgar Publishing.

2. I have some concerns regarding the fractionalisation measure proposed here. I think it is important to realise that fractionalisation is a pretty standard indicator in research on diversity generally. There are a few variants (mostly different decompositions) but they are often based on Herfindahl-Hirschmann concentration measures. Considering that the aim here is to reduce risks of false discovery, I am not quite sure why you propose your own improvement of established fractionalisation. This is not to say that I disagree with implementing new measures (to the contrary) but wouldn’t it improve comparability to previous results if you actually used the same measures? In contrast, if the intention is to contribute to the literature on measuring diversity, the proposed indicators would need to be theoretically grounded in the literature, the conceptual ideas of diversity and also analysed and documented vis-a-vis its empirical behaviour (e.g., as a first step, also descriptively). Without this, the diversity analysis here seems to be more of an empirical exercise while it could be a very relevant contribution to the literature on diversity effects in its own right. I would suggest that the authors reflect on their intended contribution and whether, given space constraints, a combination of a new methodological approach and a new indicator can be achieved here without short-changing either of these aims.

3. Following on from the previous point, I have some more concrete concerns with the proposed diversity measures.

a) It took me a few reads to understand Axiom 3. I would suggest that you re-write it for clarity. If I understand correctly, you are referring to the issue that team size affects the possible amount of diversity, i.e. it is harder for a small team to be diverse than for a large one.

b) If I understand your Axiom 3 correctly, then the fact that fractionalisation is “invariant to team (population) size by design” is an issue that is acknowledged in studies using fractionalisation or other indices (see e.g. Brixy et al., 2020). It is usually addressed by explicitly controlling for team size. Why is this not an option (or not sufficient) in your case here?

c) As far as I can see, the proposed OF index here only differs from fractionalisation because it raises the shares to the power of n (unique groups) rather than just to the power of 2, as in a HHI. Perhaps I am misunderstanding this, but it is not clear to me how this change addresses Axiom 3, since team size is not reflected in the OF equation, only number of groups. A fully diverse team of three people (i.e. three different groups) is still going to have a lower OF than a fully diverse team of four people. Could you please explain how your modification improves upon the issue of team size and how it compares to the established fractionalisation indices?

d) The notion of a composite diversity indicator is very interesting and could open interesting avenues from a conceptual perspective. However, here, almost no details on the construction of this indicator is provided, and its theoretical foundations, advantages or drawbacks are not discussed. Also, isn’t the COF just an average across fractionalisation indices of different dimensions? What does this imply for composite diversity as a concept?

4. A table showing the considered levels of the various diversity dimensions would be helpful (i.e. how many educational specialisations do you consider?) both for interpretation and also for transparency reasons (since different aggregation levels for the groups could affect the results). Also, the number and categories of control fixed effects (i.e. industry and firm size controls) would be helpful.

5. I understand that you cannot identify family-owned businesses in the data. But since this characteristic is emphasised as an explanation throughout the text, I am wondering if the authors could provide a sense of the proportion of family-owned businesses. How common are family-owned businesses in the US and in which industries? Can the authors provide any evidence supporting the claim that family-owned businesses are more likely to show higher degrees of diversity in age or gender than other types of businesses? And finally, since the authors explain the negative effects of gender and age diversity by referring to their relatively high prevalence among family-owned businesses, is it also established that these family-owned businesses are less innovative? In principle, a negative effect of gender or age diversity could be theorised: if innovation is supported by the combination of different and novel experiences, perhaps the experiences among different age groups or genders are not distinct enough to provide innovative impulses? This would of course need to be investigated further and the imprecision in the estimates remains an issue but I am wondering whether this perhaps deserves a little more detail in interpretation.

6. Depending on the intended audience, you may want to provide more detail on how the Bayesian method is implemented. The authors write (p.5) “What we used from the exploratory stage and how we arrived at reasonable values for the missing input should assist those with little experience with Bayesian estimation.” and I see that some effort is made in that respect, but I still feel that the details of the procedures could be clarified (e.g. here some (probably) trivial questions showing my ignorance, but why are the priors set with standard deviation 10? What does the traceplot look like here? Why do you use the Geweke and Heidelberger-Welch test and not the other ones?). The results are mostly clearer to follow but if this paper is indeed intended to show a protocol that can be applied rather easily, I would consider some more details on how to do that.

7. Since PLOS ONE addresses an international audience, the country of analysis should be stated clearly at least once. At the moment, the article speaks of “the Census Bureau’s Federal Statistical Research Data Center” (p. 2) and “the 2018 Annual Business Survey” but never actually specifies that we are talking about the US context here. Also, please consider whether more information on the used survey may be required (e.g. what is the coverage or sampling of this survey?) for readers unfamiliar with this dataset.

Minor points:

1. p.8: “These results, provided in Supplementary Materials, are qualitatively very similar to the de novo confirmatory estimates but represent final estimates of the effect of team diversity on innovation.” This sentence is not clear to me – what does it mean when you say that those are “final” estimates?

2. There are a few strong statements that would need to be qualified or supported with sources:

o p. 4 “Finally, previous research finds that diversity and homophily are positively associated with innovation….” needs a reference.

o p. 3 “A composite ownership fractionalization measure, which combines two or more diversity measures, represents how diversity is experienced”- I am doubtful that any quantitative measure could ever capture the actual experience of diversity.

3. In the notes for Table 1 and Table 2, the category M is missing, i.e. the only one not defined (by exclusion it seems to be education specialisation?).

4. p.8. When referring to tables 1 and 2 in the text, it would be helpful to remind the reader what the condition of inclusion in these tables was, i.e. that they show the most and least precise estimates.

6. PLOS authors have the option to publish the peer review history of their article (what does this mean?). If published, this will include your full peer review and any attached files.

Reviewer #1: No

Reviewer #2: No

Reviewer #3: No

---

## [Author Response · Author response to Decision Letter 0]

22 Jul 2024

(see attached file: Responses to reviewers.docx)

Reviewer #1

The statistical framing of this article in the abstract does not seem problematic but it is not new or a contribution. Similarly, the discussion in the introduction about the credibility crisis in applied research is also unrelated to the contribution of this article. Following best practices (or well known practices) is a necessary condition of publication, but not sufficient and not the points to lead with. I would remove both.

One of the contributions of the manuscript is the introduction of Bayesian methods to the split sample protocol for reducing the likelihood of false discovery. Because the split sample protocol imposes a significant statistical power cost on the analysis, Bayesian methods are an effective way to lessen this cost. We have not found other analyses implementing this combined split sample/dual method protocol. If such work exists, we would appreciate citations for such work and will revise our discussion accordingly.

The contribution seems to be about the causal effect of diversity of ownership teams on business innovation. I suggest leading with that.

The method introduced and demonstrated provides an implementable approach to address the credibility crisis in applied research. Accordingly, we believe this is a primary contribution to the article. While the findings concerning ownership diversity and innovation may be regarded as the main research outcome contribution, the main contribution concerning communication and public debate is related to the methods to reduce false discovery.

The paper claims that “The first step in reducing the set of estimates is to select a single measure that satisfies the requisite axioms discussed below“, but why is a single measure required? Why can’t we study multiple dimensions of this multidimensional concept? Having a single measure is convenient, but what’s the reason for this? If the separate dimensions (your axioms and other features) are correlated, you’ll only learn that if you study the dimension separately, at least at first.

Alternative measures can be proposed and used, but our introduction of the axiomatic approach to diversity measurement makes the selection of our single measure transparent. The argument against multiple measures applied to 7 different diversity dimensions is that the “researcher degrees of freedom” quickly becomes very large. Moreover, the probability of false discovery increases as research degrees of freedom increase. The research approach we are arguing against is the nontransparent investigation of a slew of alternative measures that result in the “preferred” measure being published, accompanied by a post hoc justification. All possible results are estimated and reported by constructing a single measure that satisfies explicit axioms. A single diversity measure addresses the multiple comparison problem by facilitating false discovery rate and family-wise error rate adjustments. There may be conceptual disagreements with the validity of our measure, and we would encourage other researchers to develop and test alternative measures if the same approach to transparency is followed. What critical axioms does our measure miss? Why are our axioms that the new measure does not satisfy invalid? These debates can flourish under an axiomatic approach. However, productive debate is much less efficient if diversity measurement searches are part of unreported specification searches.

If you do have one dimension, and we try to satisfy the (reasonable) axioms proposed, then the authors claim that they have a measure proposed satisfies them. This is merely stated. It seems true, but it needs to be proven. But more importantly, is this the unique measure that satisfies the axioms? A proof that this is unique would be useful. (A somewhat similar example of a uniqueness proof, done by building on entropy, is in doi:10.1093/pan/mpl011). If not, then do the empirical results vary depending on which measure (among those that fit the axioms) is used?

An empirical demonstration of how the diversity index changes with changes in the number of owners, the number of unique groups, and ownership concentration has been added to the manuscript on page 7. A comparison with the commonly used ethnolinguistic fractionalization measure is also provided. 

The 2018 *annual* business survey seems like a useful dataset, but why stop in 2018? The article has the laudable training/test framework, and so why not -- following that setup -- use a purely out of sample test set with one of the subsequent years of data?

The main reason for limiting the analysis to a single dataset is that this is the situation that most researchers interested in implementing a split sample/dual method protocol will be in. In addition, the 2018 ABS (reference year 2017) is more than twice as large as the following years (2019-2022), given the Census Bureau’s interest in having more information available in the Economic Census years. Finally, using observations from the same year eliminates one possible explanation for qualitative differences between exploratory and confirmatory tests due to temporal shifts in the phenomenon. All the external factors are the same, so change in the phenomenon over time is not a possible explanation. 

A causal identification assumption is required to make [Disp-formula pone.0313826.e001] useful. This needs to be clearly stated and justified. I’m not sure that is possible here, but it at least needs to be addressed. It is the central point of the entire paper.

No claims concerning causality are made in the manuscript. The analysis is useful because it provides transparent evidence of the association between ownership diversity and innovation. This objective is clarified on page 2, and the limitation is identified in the Discussion on page 17. A causal analysis would be required to support claims that increasing diversity of ownership teams will increase innovation.

[Disp-formula pone.0313826.e001] also makes big homogeneity assumptions (such as beta1 not varying by the fixed effects). These should be tested and explored. I would suggest that the authors first run their analysis separately within different firm size and industry classes and then only put them together into one equation if the data support that. More likely there are different types of effects in different industries or types of firms, but this will be apparent if you look rather than assume.

Including industry and firm size fixed effects in microdata analysis is a widely accepted procedure for controlling possible confounds of the relationship interest (see Akcigit, U. and Goldschlag, N., 2023. Measuring the characteristics and employment dynamics of U.S. inventors (No. w31086). National Bureau of Economic Research doi:10.3386/w31086, as one of many examples). The homogeneity assumption of differences in the likelihood of reporting innovation being expressed as a shift in the intercept that differs by industry or firm size is not particularly heroic and is well supported by differences in average innovation rates across industry and firm size. The suggested alternative to do numerous subsample analyses decreases statistical power, increases researcher degrees of freedom by increasing the number of separate analyses, and substantially increases the multiple comparison problem. The Bayesian analysis suggests that a mixed model that uses fixed effects for industry and random effects for firm size provides better estimates, passing diagnostic tests that fail in some estimations when only fixed effects are used. The assumption of fixed effects for industry and firm size that poses no problems for the frequentist analysis does run into problems in the Markov chain Monte Carlo analysis that may be due to a poorly specified model. 

The “registered report” framework here is hurting the analysis. The authors are not learning from the data and so the readers are not learning from the authors. I would jettison that report and just do a proper model-free statistical exploration.

We view the split sample/dual method protocol as a middle ground between pre-analysis plans where no learning from the data is allowed and model-free statistical exploration that characterizes the data-dependent analysis that predominates in the social sciences. The structure of our protocol only eliminates learning from the data in the confirmatory stage. More specification testing could have been done in the exploratory stage but was limited, given an interest in simplifying the proof of concept. The important dividend of de novo estimation in the confirmatory stage is that the validity of hypothesis test statistics is preserved. Ensuring that the nominal p-value reported in publications are true p-values is the first step in addressing the credibility crisis in the social sciences.

Reviewer #2

The manuscript demonstrates a robust and technically sound methodology, utilizing both frequentist and Bayesian approaches to enhance the reliability of findings. The use of a split-sample/dual-method protocol is a novel approach that enhances research transparency and addresses issues of reproducibility. The study addresses a highly relevant and contentious topic with significant implications for diversity and innovation policies. However, there are still few comments and suggestions for improvement as follow:

(1) Complexity of Methodological Explanation: The explanation of the dual-method approach and the statistical techniques used, including Bayesian methods and corrections for multiple comparisons, is highly technical and may be challenging for readers without a strong statistical background. Therefore, it is advisable to Simplify the methodological sections by providing a more straightforward summary or an appendix with detailed explanations and step-by-step guides. Including visual aids such as diagrams to illustrate the research design and statistical methods could also be beneficial.

The multiple comparison problem is better illustrated by explicitly defining the p-value and how the 127 different estimates would make conventional hypothesis tests uninformative. This discussion is added on page 7, starting on line 239. One of the primary objectives of the article is to demonstrate the potential benefits of methodological pluralism, so the encouragement to lay out the protocol more simply is appreciated. We have produced a table outlining the distinct steps in both the previous exploratory analysis and the current confirmatory analysis which is now included in Supplementary Materials. We have also expanded the discussion regarding specifying and estimating Bayesian models that are based on frequentist estimation and have provided more references geared to researchers making their first foray into Bayesian analysis. We include SAS code in the Supplemental Materials for interested readers (Table S5). 

(2) More Discussion on Practical Implications is needed: The manuscript provides an extensive methodological discussion but lacks depth in discussing the practical implications of the findings for business practices and policy-making, authors are advised to explain further how findings could influence business strategies, diversity policies, and innovation practices.

One of the authors is a government employee, so there are some limits to the discussion about policy implications or actions that might be taken, given the findings. The major limitation of the analysis for informing the practical implications of the diversity/innovation association is that causality is not established. (This was not the objective of the manuscript). We discuss this limitation in the final section on page 17, beginning on line 537. This analysis focused on providing the most transparent and robust estimates of the association as an impetus for research that plausibly controls for endogeneity and is extended to innovation outcomes mediated in markets. These issues are added to the end of the concluding Discussion section beginning on line 539.

(3) Proofreading Needs, a final proofread to catch minor typographical errors and ensure consistency in terminology and formatting would improve the manuscript’s polish.

Revisions from line editing and proof reading are indicated in track changes.

(4) Two Self-Citation were found, While self-citations are acceptable, ensuring they do not dominate the reference list

Because research using the self-reported innovation questions in the ABS is still quite limited, self-citations are sometimes necessary. Material first developed in the registered report has been added to this manuscript to address comments from reviewers requesting elaboration on the ownership diversity measure.

Reviewer #3: Dear authors,

this is a very interesting paper pursuing several ambitious and relevant goals. I absolutely agree with the need for increased research transparency in social sciences/economics and appreciate the authors’ engagement with these issues and the proposed ideas and protocol. However, considering the different aims and contributions made, the paper would benefit from a clearer focus. This includes, on the one hand, engaging a little more with the existing literature on innovative effects of diversity and, on the other hand, ensuring that the proposed protocol is accessible for the intended audience.

Major points

1. I understand that the conceptual and empirical background on diversity effects is not the focus of this paper and is therefore kept short. However, as an application addressing (in the authors’ words) a “contentious” issue, and certainly one attracting both theoretical, empirical and policy attention, I find the engagement with the literature superficial. I broadly agree with the theoretical reasoning for both innovative advantages and hinderances but would suggest to acknowledge the existing literature on diversity effects a bit more explicitly. This applies especially to the section “Diversity Findings Vulnerable to False Discovery” but also to the sections discussing the results. Some suggestions:

o for an overview of the literature: Ozgen, C. (2021). The economics of diversity: Innovation, productivity and the labour market. Journal of Economic Surveys, 35(4), 1168–1216. https://doi.org/10.1111/joes.12433

o for a specific application of team diversity to firm innovation: Brixy, U., Brunow, S., & D’Ambrosio, A. (2020). The unlikely encounter: Is ethnic diversity in start-ups associated with innovation? Research Policy, 49(4), 103950. https://doi.org/10.1016/j.respol.2020.103950

o as a seminal reference for the existence of diversity effects: Page, S. E. (2007). The difference: How the power of diversity creates better groups, firms, schools, and societies. Princeton Univ. Press.

o for a review of different measures of diversity: : Nijkamp, P., & Poot, J. (2015). Cultural diversity: A matter of measurement. In The Economics of Cultural Diversity (pp. 17–51). Edward Elgar Publishing.

A fuller discussion from the literature of how diversity may promote or hinder innovation has been added on pages 2-3. A reference to Nijkamp & Poot is very helpful for demonstrating the index search problem that might contribute to false discovery.

I have some concerns regarding the fractionalisation measure proposed here. I think it is important to realise that fractionalisation is a pretty standard indicator in research on diversity generally. There are a few variants (mostly different decompositions) but they are often based on Herfindahl-Hirschmann concentration measures. Considering that the aim here is to reduce risks of false discovery, I am not quite sure why you propose your own improvement of established fractionalisation. This is not to say that I disagree with implementing new measures (to the contrary) but wouldn’t it improve comparability to previous results if you actually used the same measures? In contrast, if the intention is to contribute to the literature on measuring diversity, the proposed indicators would need to be theoretically grounded in the literature, the conceptual ideas of diversity and also analysed and documented vis-a-vis its empirical behaviour (e.g., as a first step, also descriptively). Without this, the diversity analysis here seems to be more of an empirical exercise while it could be a very

---

## [Decision Letter · Decision Letter 1]

6 Sep 2024

PONE-D-24-08785R1Research Transparency on Contentious Topics: Exploring the Connection between Diversity and InnovationPLOS ONE

Dear Dr. Wojan,

Thank you for submitting your manuscript to PLOS ONE. After careful consideration, we feel that it has merit but does not fully meet PLOS ONE’s publication criteria as it currently stands. Therefore, we invite you to submit a revised version of the manuscript that addresses the points raised during the review process.

**Thank you for your valuable submission.**

1) Please double-check grammar (e.g. punctuation and verb tense);

2) Please double-check refs (e.g. abbreviations, page numbers, and sentence case need to be corrected);

3) Although the title is not concerning, I believe the authors could improve by focusing on a specific aspect of the innovation. For example, Bayesian estimation is correct, but for a broad audience, it won't be as strong as "novel framework" or even "Robust methods for...". These are only simple examples requesting authors to make it more 'punchy' yet important and clear for researchers too;

4) Consider refining the Abstract, particularly in terms of language and the presentation of information (e.g. "Bayesian estimation replaces traditional [...] in the confirmatory stage" seems a little bit more direct and strong statement). The authors are strongly encouraged to make the Abstract more engaging, emphasising the novelty and relevance of the research;

- Complex terms such as 'priors' and 'posteriors' could perhaps be introduced in the Introduction rather than the Abstract to make the latter more accessible;

- If the options for Title are not particularly engaging, consider rephrasing terms like 'Bayesian methods' to make it engaging for both experts and the broad audience. This also applies to the wording in the Abstract;

- Please work on a strong introductory line that emphasises the relevance of the study;

- Provide consistent results based on your study with its parameters;

- Additionally, strengthen the Abstract reinforcing the importance of the findings, and make the practical implications of the study more explicit;

5) Simplify sections where the focus shifts towards stats and use clearer language to explain complex concepts;

6) The Introduction needs to provide a clearer explanation of how the chosen diversity measure aligns with the stated axioms and why it was selected over alternatives;

7) The absence of a Methods section or, at the very least, an in-depth explanation of search, data interpretation and presentation focussing on stats is a little bit worrying. I'd highly encourage the authors to reshape the sections preceding the Tables, focusing on explaining how the data was gathered and how it will be displayed in terms of stats parameters. A sup section could also be important;

8) The presentation of OF, EF, and COF is commendable, though the text could benefit from further refinement. Specifically, it could elaborate on: a) how the indices interact with the different dimensions; and b) a clearer and more direct argument (with data from both simulations and practical examples) on how Bayesian methods can help mitigate power loss in frequentist models;

- The introduction of the OF and COF indices is innovative, but it would benefit from a more explicit explanation of how they compare to traditional diversity indices beyond the tables. For instance, explaining why the exploratory stage mitigates concerns about subjective priors could help readers;

9) Consider tidying the Tables;

10) Overall, the Tables are dense with numerous parameters. The abbreviations are unclear without constant reference to the footnotes;

- Not all readers require both Frequentist and Bayesian estimates to be presented simultaneously. Streamlining the Tables prioritising the most significant data or presenting detailed Bayesian diagnostics separately would enhance clarity;

- For the Frequentist model, providing R-squared or adjusted R-squared values would help evaluate how well the model fits the data;

- It is unclear at a glance which estimates are meaningful, given the numerous values and parameters presented;

- While it may be implicit, the use of Forest Plots could be particularly useful in simplifying the density and presentation of the data;

- The same for presentation of priors - or mention before the Tables;

11) The authors are encouraged to use more references while debating and presenting the arguments in Discussion;

- It'd be valuable to include more refs and explanations on the prevalence of HARKing, claims regarding irreproducible studies, and in-depth references for specific information, such as in line 526: "climate change, public health, diversity";

- What 'Registered Report'?;

- Are other studies or meta-analyses relating to diversity and innovation? If so, they should be mentioned and explored;

In addition, consider updating refs, including those by Nosek, Baltagi, or Vehtari, for example. Also, strengthen your argument by citing relevant literature, and provide a clearer summary of the changes made to the ms.

Wishing you success with the study.

We look forward to receiving your revised manuscript.

Kind regards,

Thiago P. Fernandes, PhD

Academic Editor

PLOS ONE

Reviewers' comments:

Reviewer's Responses to Questions

**Comments to the Author**

1. If the authors have adequately addressed your comments raised in a previous round of review and you feel that this manuscript is now acceptable for publication, you may indicate that here to bypass the “Comments to the Author” section, enter your conflict of interest statement in the “Confidential to Editor” section, and submit your "Accept" recommendation.

Reviewer #3: (No Response)

Reviewer #4: All comments have been addressed

Reviewer #5: All comments have been addressed

2. Is the manuscript technically sound, and do the data support the conclusions?

Reviewer #3: Partly

Reviewer #4: Yes

Reviewer #5: No

3. Has the statistical analysis been performed appropriately and rigorously? 

Reviewer #3: Yes

Reviewer #4: No

Reviewer #5: No

4. Have the authors made all data underlying the findings in their manuscript fully available?

Reviewer #3: Yes

Reviewer #4: (No Response)

Reviewer #5: No

5. Is the manuscript presented in an intelligible fashion and written in standard English?

Reviewer #3: Yes

Reviewer #4: Yes

Reviewer #5: No

6. Review Comments to the Author

**Reviewer #3: **Dear Authors,

Thank you for the opportunity to read the revised version of this manuscript. I believe that the paper has much improved and I appreciate the work that the authors have invested. In particularly, the justification for the analysis and the detail of the method description are both much clearer now.

However, I still have concerns with the proposed measure of ownership fractionalisation (OF). While the justification for the axioms is clearer now, there is still no actual engagement with the conceptual meaning of this indicator or even proof that it satisfies the axioms. The “simple modifications” suggested to ensure that the axioms are satisfied are not conceptually explained or justified and seem to be derived purely mathematically, whereas the EF does have a conceptual grounding. This leads to a strange comparison of a well-known and conceptually valuable index with an almost arbitrary-seeming mathematical construct.

The authors instead provide two tables that are supposed to compare the OF with the established fractionalisation indicator based on an HHI. I see two issues with these tables:

1. An easy to fix but important problem is that the rows and columns in table 2 and table 3 seem to be mislabeled (or flipped). It says that the rows are number of owners and columns are unique groups but this does not correspond to the content of the tables. As it is presented now, the table should show OFs for two owners (row 1) across two, three and four unique groups, which would mean that owners can be in more than one group simultaneously. While this is of course possible theoretically, it falls outside the proposed OF indicator (and is rather an example of the COF or other multidimensional measures). In fact, when tracing the numbers in table 2, it is clear that the numbers in the last column (4-equally distributed) correspond to 1 – 4*((1/4)^2) = 0.75, 1 – 4*((1/4)^3) = 0.9375 and 1- 4*((1/4)^4)=0.9844. According to equation 1, the exponent n identifies the number of groups, not the number of owners as the table now suggests. It seems that the labels have been flipped in the tables without adjusting the numbers.

2. I also do not agree with the calculations in table 3. Clearly, the EF does not have a “number of owners” dimension but is only calculated for the number of groups. I understand that this is what you are trying to show by repeating the same value in the table but I think that this is an oversimplification and the values in the table are actually not entirely correct or not clearly presented.

Because the EF is calculated based on the relative prevalence of the unique groups (not based on ownership shares) changing the number of owners while keeping the number of groups constant would in fact change the EF value in some cases. Consider the case of 2 unique groups: it is true that the value for two owners and four owners with equal distribution would both be 0.5. But for three owners, the EF would actually be 1-((1/3)^2+(2/3)^2)=0.6667 because the distribution across groups would necessarily be unequal. Since the EF depends on the relative proportions of the groups as determined by both the number of groups and the number of owners, it is actually not entirely “invariant” to team size. The same applies for three unique groups and three versus four owners.

Moreover, the columns using “concentrated” ownership do not really makes sense for the EF since they are not formally captured. Of course, we can just use ownership shares analogously to how we would use population shares here but then the EF-value for some constellations still depends on the distribution across groups. For instance, a team with two unique groups and three owners and concentrated shares could either have an EF of 1 – (0.01^2+0.99^2)=0.0198 (just like a team with two owners) or of 1-(0.02^2+0.98^2)=0.0392. Without more information on the concentration across the two groups, either of these values could occur.

In essence: the example comparisons the authors present in table 3 do not actually capture the behaviour of the EF as it is not as static as suggested here but does in fact respond to changes in team size as long as these have an effect on relative prevalence of the groups. Thus, the empirical demonstration that was meant to clarify the behaviour of the OF relative to the EF thus does not help to clarify the added value and/or distinction between the two indices yet but rather highlights the complexity of these measures.

I would suggest either revising the tables to show the behaviour of the two indices more comprehensively and correctly (perhaps with a less abstract example?) or perhaps also consider other methods of comparison. For instance, I wonder (just out of curiosity) about the simple correlations between the EF and OF measures in your data: Do they tend in the same direction on average? How different are these two indices really?

**Reviewer #4:** "I am pleased to announce that all the issues have been thoroughly addressed as desired, and the work is now in excellent shape, ready for publication. Through careful revisions and attention to detail, all necessary enhancements have been implemented, ensuring the quality and accuracy of the content. The result reflects a refined and polished piece, meeting the highest standards of excellence. I am confident that this publication will contribute significantly to the field and resonate well with the target audience. I'm excited to share this finalized work and look forward to the positive impact it will have."

**Reviewer #5: **After a careful study of the article, reviewers' comments and corrections, I have come to the conclusion that the quality of the manuscript has not improved and fundamental problems including the theoretical contribution of the research can be seen in the literature.

7. PLOS authors have the option to publish the peer review history of their article (what does this mean?). If published, this will include your full peer review and any attached files.

Reviewer #3: No

Reviewer #4: **Yes: **Muhammad Zunnurain Hussain

Reviewer #5: No

---

## [Author Response · Author response to Decision Letter 1]

2 Oct 2024

Thank you for your valuable submission.

1) Please double-check grammar (e.g. punctuation and verb tense);

In track changes.

2) Please double-check refs (e.g. abbreviations, page numbers, and sentence case need to be corrected);

Checked. Page dashes made consistent with ICMJE. Sentence case for titles with colons now conforms with “Vancouver” style, as outlined in the ICMJE.

3) Although the title is not concerning, I believe the authors could improve by focusing on a specific aspect of the innovation. For example, Bayesian estimation is correct, but for a broad audience, it won't be as strong as "novel framework" or even "Robust methods for...". These are only simple examples requesting authors to make it more 'punchy' yet important and clear for researchers too;

Thanks for the suggestion. “A novel framework for increasing research transparency: exploring the connection between diversity and innovation” is punchier and avoids the contentious topic redundancy that should be clear from the subject matter. 

4) Consider refining the Abstract, particularly in terms of language and the presentation of information (e.g. "Bayesian estimation replaces traditional [...] in the confirmatory stage" seems a little bit more direct and strong statement). The authors are strongly encouraged to make the Abstract more engaging, emphasising the novelty and relevance of the research;

- Complex terms such as 'priors' and 'posteriors' could perhaps be introduced in the Introduction rather than the Abstract to make the latter more accessible;

- If the options for Title are not particularly engaging, consider rephrasing terms like 'Bayesian methods' to make it engaging for both experts and the broad audience. This also applies to the wording in the Abstract;

- Please work on a strong introductory line that emphasises the relevance of the study;

- Provide consistent results based on your study with its parameters;

- Additionally, strengthen the Abstract reinforcing the importance of the findings, and make the practical implications of the study more explicit;

We see the main contribution of this paper as being methodological: demonstrating a framework that can directly address the credibility crisis in applied research. Accordingly, we retain the original first sentence of the abstract. We have removed more technical terms from the abstract and streamlined the description of the protocol. The magnitude of the impact of diversity is now included in the abstract but we are reticent to outline practical implications of the study as are findings are correlational rather than causal.

5) Simplify sections where the focus shifts towards stats and use clearer language to explain complex concepts;

We have examined these sections and believe that they provide readers using statistical techniques in their own research adequate information to understand what was done.

6) The Introduction needs to provide a clearer explanation of how the chosen diversity measure aligns with the stated axioms and why it was selected over alternatives;

The introduction now discusses the axiomatic selection of a diversity measure to reduce researcher degrees of freedom. A fuller discussion of how the ownership fractionalization measure satisfies these axioms has been added to the second section on “Diversity findings vulnerable to false discovery.” 

7) The absence of a Methods section or, at the very least, an in-depth explanation of search, data interpretation and presentation focussing on stats is a little bit worrying. I'd highly encourage the authors to reshape the sections preceding the Tables, focusing on explaining how the data was gathered and how it will be displayed in terms of stats parameters. A sup section could also be important;

The discussion in the section “Specification of the innovation and ownership diversity regression equation” provides the justification and interpretation of the variables used in the analysis that are all provided in the Annual Business Survey. An explanation of the statistics to be provided in the confirmatory results tables is provided in the “Exploratory results” section that defines the FWER and FDR corrections to be used for the confirmatory results.

8) The presentation of OF, EF, and COF is commendable, though the text could benefit from further refinement. Specifically, it could elaborate on: a) how the indices interact with the different dimensions; and b) a clearer and more direct argument (with data from both simulations and practical examples) on how Bayesian methods can help mitigate power loss in frequentist models;

In response to comments from Reviewer 3 we discuss more fully the logic behind the 4 axioms and how the OF/COF satisfies them. This has allowed us to simplify the EF/OF comparison tables that are now limited to maximal values for different values of the number of groups and number of owners. The demonstration of ownership concentration in the previous tables was ambiguous and has been removed. We also estimated the correlation of the EF and OF indices which is very high (>0.90) for dimension with 2 groups, weakening somewhat as the number of groups in a dimension increase. Results from using the EF index in place of the OF index are qualitatively similar but of smaller magnitude and this is mentioned in the manuscript on page 22. 

- The introduction of the OF and COF indices is innovative, but it would benefit from a more explicit explanation of how they compare to traditional diversity indices beyond the tables. For instance, explaining why the exploratory stage mitigates concerns about subjective priors could help readers;

The expanded discussion of how the OF satisfies the four axioms should make the measure selection more persuasive to readers. In addition, we have examined the sensitivity of our findings to replacing OF with EF and the results are qualitatively similar, with the two measures being strongly correlated. The main difference between OF and EF is the sensitivity to team size which we argue is essential in the study of small teams where interaction among members is guaranteed. This discussion begins on line 177. 

9) Consider tidying the Tables;

Many of the statistics presented in Tables 4 and 5 may appear superfluous relative to reporting as commonly done in journal articles, particularly with respect to FDR, FWER (Bonferroni correction) and Bayesian credible intervals. However, all of this information is essential for fully understanding the implications of the multiple comparison problem and assessing the possible advantages of methodological pluralism in this proof-of-concept application. Subsequent papers using the split sample/dual method protocol will be able to present tidier tables only including the Bayesian estimates while referencing this paper to establish advantages with respect to statistical power.

10) Overall, the Tables are dense with numerous parameters. The abbreviations are unclear without constant reference to the footnotes;

Abbreviations have been added after the variable name labels in Table 6 that should aid readers scanning the document who may not be looking at table footnotes. Naming COF variables with the diversity dimension labels becomes intractable when the number of diversity dimensions exceeds 3 or 4.

- Not all readers require both Frequentist and Bayesian estimates to be presented simultaneously. Streamlining the Tables prioritising the most significant data or presenting detailed Bayesian diagnostics separately would enhance clarity;

Demonstrating the advantages to methodological pluralism is a central objective of the manuscript. Simultaneous presentation of the frequentist and Bayesian estimates accomplishes this. In addition to the demonstration of improved power in the Bayesian estimation, side-by-side estimates may allay concerns from readers unfamiliar with Bayesian methods that the weakly informative prior does not “put a thumb on the scale” of the Bayesian estimate. While this may be a concern for much smaller samples, the side-by-side comparison demonstrates that likelihood dominates with large samples.

- For the Frequentist model, providing R-squared or adjusted R-squared values would help evaluate how well the model fits the data;

R-squared in large micro datasets with parsimonious models of complex processes like innovation often provides little basis for evaluation of model fit. It is not included given the large number of statistics in the tables.

- It is unclear at a glance which estimates are meaningful, given the numerous values and parameters presented;

The Bonferroni confidence intervals and equal-tailed credible intervals are now bold in Table 4 and 5 as these are the estimates that are most meaningful.

- While it may be implicit, the use of Forest Plots could be particularly useful in simplifying the density and presentation of the data;

The objective of presenting a large number of diversity estimates is not to rank or compare them necessarily but rather to provide assurances to the reader that the results have not been cherry-picked. A visual display of credible intervals could reinforce a ranking objective that is not the intent of the manuscript.

- The same for presentation of priors - or mention before the Tables;

Inclusion of the priors on Tables 4 and 5 serves several purposes: 1) it makes clear that the prior is an input in the Bayesian estimation; 2) it provides a link back to the exploratory analysis used in the split sample/dual method protocol; and 3) it demonstrates to readers that weakly informed priors in estimations using large datasets have little effect on the final Bayesian estimates other than shrinking the standard deviation in the posterior distribution. 

11) The authors are encouraged to use more references while debating and presenting the arguments in Discussion;

- It'd be valuable to include more refs and explanations on the prevalence of HARKing, claims regarding irreproducible studies, and in-depth references for specific information, such as in line 526: "climate change, public health, diversity";

References supporting the Discussion section have been expanded considerably. In addition, references cited earlier in the paper are added where appropriate to demonstrate how the current research addresses the problems of data dependent analysis.

- What 'Registered Report'?; 

[27] from the References now included in the text.

- Are other studies or meta-analyses relating to diversity and innovation? If so, they should be mentioned and explored;

We have been unable to find any studies or meta-analyses of ownership diversity and innovation. There are studies of diversity in the workplace but there are very few data sources like the Annual Business Survey that combine an ownership survey (the former Survey of Business Owners) and an innovation survey (the innovation module from the former Business R&D and Innovation Survey). Among OECD countries that follow the Oslo Manual for innovation surveys, the U.S. is the only country that includes information on company owners.

In addition, consider updating refs, including those by Nosek, Baltagi, or Vehtari, for example. Also, strengthen your argument by citing relevant literature, and provide a clearer summary of the changes made to the ms.

Reference to Nosek added along with other references in Discussion section that reinforce our argument.

Wishing you success with the study.

Reviewer #3: Dear Authors,

Thank you for the opportunity to read the revised version of this manuscript. I believe that the paper has much improved and I appreciate the work that the authors have invested. In particularly, the justification for the analysis and the detail of the method description are both much clearer now.

However, I still have concerns with the proposed measure of ownership fractionalisation (OF). While the justification for the axioms is clearer now, there is still no actual engagement with the conceptual meaning of this indicator or even proof that it satisfies the axioms. The “simple modifications” suggested to ensure that the axioms are satisfied are not conceptually explained or justified and seem to be derived purely mathematically, whereas the EF does have a conceptual grounding. This leads to a strange comparison of a well-known and conceptually valuable index with an almost arbitrary-seeming mathematical construct.

The authors instead provide two tables that are supposed to compare the OF with the established fractionalisation indicator based on an HHI. I see two issues with these tables:

1. An easy to fix but important problem is that the rows and columns in table 2 and table 3 seem to be mislabeled (or flipped). It says that the rows are number of owners and columns are unique groups but this does not correspond to the content of the tables. As it is presented now, the table should show OFs for two owners (row 1) across two, three and four unique groups, which would mean that owners can be in more than one group simultaneously. While this is of course possible theoretically, it falls outside the proposed OF indicator (and is rather an example of the COF or other multidimensional measures). In fact, when tracing the numbers in table 2, it is clear that the numbers in the last column (4-equally distributed) correspond to 1 – 4*((1/4)^2) = 0.75, 1 – 4*((1/4)^3) = 0.9375 and 1- 4*((1/4)^4)=0.9844. According to equation 1, the exponent n identifies the number of groups, not the number of owners as the table now suggests. It seems that the labels have been flipped in the tables without adjusting the numbers.

The original Tables 2 and 3 have been replaced with maximal value tables (justification below) with the corrected row and column labelling.

2. I also do not agree with the calculations in table 3. Clearly, the EF does not have a “number of owners” dimension but is only calculated for the number of groups. I understand that this is what you are trying to show by repeating the same value in the table but I think that this is an oversimplification and the values in the table are actually not entirely correct or not clearly presented.

Because the EF is calculated based on the relative prevalence of the unique groups (not based on ownership shares) changing the number of owners while keeping the number of groups constant would in fact change the EF value in some cases. Consider the case of 2 unique groups: it is true that the value for two owners and four owners with equal distribution would both be 0.5. But for three owners, the EF would actually be 1-((1/3)^2+(2/3)^2)=0.6667 because the distribution across groups would necessarily be unequal. Since the EF depends on the relative proportions of the groups as determined by both the number of groups and the number of owners, it is actually not entirely “invariant” to team size. The same applies for three unique groups and three versus four owners.

EF in the literature is used with populations and the resulting population shares across groups, not with small teams with a maximum number of 4. However, applying EF to small teams can result in the anomalies of the measure being sensitive to team size as noted. To address this issue along with the ambiguities related to ownership concentration we have replaced the original tables with maximal value tables. These tables demonstrate the differences in possible range of the indices given changes in the number of groups and number of owners. 

Moreover, the columns using “concentrated” ownership do not really makes sense for the EF since they are not formally captured. Of course, we can just use ownership shares analogously to how we would use population shares here but then the EF-value for some constellations still depends on the distribution across groups. For instance, a team with two unique groups and three owners and concentrated shares could either have an EF of 1 – (0.01^2+0.99^2)=0.0198 (just like a team with two owners) or of 1-(0.02^2+0.98^2)=0.0392. Without more information on the concentration across the two groups, either of these values could occur.

Ownership concentration has been removed from the EF and OF comparison tables to address the ambiguities of distribution across groups

---

## [Decision Letter · Decision Letter 2]

1 Nov 2024

A Novel Framework for Increasing Research Research Transparency: Exploring the Connection between Diversity and Innovation

PONE-D-24-08785R2

Dear Dr. Wojan,

We’re pleased to inform you that your manuscript has been judged scientifically suitable for publication and will be formally accepted for publication once it meets all outstanding technical requirements.

Kind regards,

Thiago P. Fernandes, PhD

Academic Editor

PLOS ONE

Additional Editor Comments (optional):

Reviewers' comments:

Reviewer's Responses to Questions

**Comments to the Author**

1. If the authors have adequately addressed your comments raised in a previous round of review and you feel that this manuscript is now acceptable for publication, you may indicate that here to bypass the “Comments to the Author” section, enter your conflict of interest statement in the “Confidential to Editor” section, and submit your "Accept" recommendation.

Reviewer #3: All comments have been addressed

2. Is the manuscript technically sound, and do the data support the conclusions?

Reviewer #3: Yes

3. Has the statistical analysis been performed appropriately and rigorously? 

Reviewer #3: Yes

4. Have the authors made all data underlying the findings in their manuscript fully available?

Reviewer #3: (No Response)

5. Is the manuscript presented in an intelligible fashion and written in standard English?

Reviewer #3: Yes

6. Review Comments to the Author

Reviewer #3: (No Response)

7. PLOS authors have the option to publish the peer review history of their article (what does this mean?). If published, this will include your full peer review and any attached files.

Reviewer #3: No

---

## [Editor Report · Acceptance letter]

7 Nov 2024

PONE-D-24-08785R2 

PLOS ONE

Dear Dr. Wojan, 

I'm pleased to inform you that your manuscript has been deemed suitable for publication in PLOS ONE. Congratulations! Your manuscript is now being handed over to our production team.

Kind regards, 

on behalf of

Dr. Thiago P. Fernandes 

Academic Editor

PLOS ONE